# The rs368698783 (G>A) Polymorphism Affecting LYAR Binding to the *Aγ-Globin* Gene Is Associated with High Fetal Hemoglobin (HbF) in β-Thalassemia Erythroid Precursor Cells Treated with HbF Inducers

**DOI:** 10.3390/ijms24010776

**Published:** 2023-01-01

**Authors:** Cristina Zuccato, Lucia Carmela Cosenza, Matteo Zurlo, Giulia Breveglieri, Nicoletta Bianchi, Ilaria Lampronti, Jessica Gasparello, Chiara Scapoli, Monica Borgatti, Alessia Finotti, Roberto Gambari

**Affiliations:** 1Section of Biochemistry and Molecular Biology, Department of Life Sciences and Biotechnology, University of Ferrara, 44121 Ferrara, Italy; 2Center ‘Chiara Gemmo and Elio Zago’ for the Research on Thalassemia, University of Ferrara, 44121 Ferrara, Italy; 3Department of Translational Medicine and for Romagna, University of Ferrara, 44121 Ferrara, Italy; 4Section of Evolutionary Biology, Department of Life Sciences and Biotechnology, University of Ferrara, 44121 Ferrara, Italy

**Keywords:** β-thalassemia, fetal hemoglobin, rs368698783, LYAR

## Abstract

The human homologue of mouse Ly-1 antibody reactive clone protein (LYAR) is a putative novel regulator of *γ-globin* gene transcription. The LYAR DNA-binding motif (5′-GGTTAT-3′) is located within the 5′-UTR of the *Aγ-globin* gene. The LYAR rs368698783 (G>A) polymorphism is present in β-thalassemia patients and decreases the LYAR binding efficiency to the *Aγ-globin* gene. The objective of this study was to stratify β-thalassemia patients with respect to the rs368698783 (G>A) polymorphism and to verify whether their erythroid precursor cells (ErPCs) differentially respond in vitro to selected fetal hemoglobin (HbF) inducers. The rs368698783 (G>A) polymorphism was detected by DNA sequencing, hemoglobin production by HPLC, and accumulation of globin mRNAs by RT-qPCR. We found that the LYAR rs368698783 (G>A) polymorphism is associated with high basal and induced production of fetal hemoglobin in β-thalassemia patients. The most striking association was found using rapamycin as an HbF inducer. The results presented here could be considered important not only for basic biomedicine but also in applied translational research for precision medicine in personalized therapy of β-thalassemia. Accordingly, our data suggest that the rs368698783 polymorphism might be considered among the parameters useful to recruit patients with the highest probability of responding to in vivo hydroxyurea (HU) treatment.

## 1. Introduction

The β-thalassemias are impactful hereditary hematological diseases caused by nearly 300 mutations of the *β-globin* gene leading to low or absent production of adult hemoglobin (HbA) [1,2,3,4]. An increased expression of *γ-globin* genes and high fetal hemoglobin (HbF) production in β-thalassemia patients are associated with a milder or even asymptomatic disease pattern [5,6,7]. In this case, a transfusion regimen and chelation therapy might not be necessary [2,3,5]. Therefore, efficient HbF inducers were developed for the treatment of β-thalassemia patients characterized by low production of HbF [5,7,8,9,10,11]. Since the response to HbF inducers might depend on HbF-associated polymorphisms (such as the XmnI, BCL11A, and MYB polymorphisms) [4,5], patient stratification based on these genetic characteristics might be useful for clinical management and choice of the therapeutic protocol [12,13].

Recently, Ju et al. [14] identified a putative novel regulator of *γ-globin* gene transcription, LYAR (human homologue of mouse Ly-1 antibody reactive clone). The LYAR DNA-binding motif (5′-GGTTAT-3′) was identified in a DNA region corresponding to the 5′-untranslated region of the *Aγ-globin* gene and to the rs368698783 (G>A) polymorphism. LYAR is a strong repressor of human *γ-globin* gene transcription, suggesting this is a novel transcription factor essential for *γ-globin* gene silencing [14]. Interestingly, we found that the rs368698783 (G>A) polymorphism is present in β-thalassemia patients [15,16]. Moreover, molecular docking simulations, an electrophoretic mobility shift assay (EMSA), and Biacore analyses demonstrated that the rs368698783 (G>A) polymorphism decreases the LYAR binding efficiency to the *Aγ-globin* gene, suggesting that it might affect cellular responses to HbF inducers [15,17].

The objective of the present study was to stratify β-thalassemia patients with respect to the rs368698783 (G>A) polymorphism and to verify whether this polymorphism influences the in vitro response to selected HbF inducers. As in vitro cellular model system erythroid precursor cells (ErPCs) from β-thalassemia patients were employed [18], the HbF inducers considered in this study were mithramycin (MTH) [19], hydroxyurea (HU) [20], and rapamycin (RAPA) [21,22,23].

## 2. Results

### 2.1. Heterogeneity of the HbF Induction of ErPCs from Different β-Thalassemia Patients

The Location of the rs368698783 (G>A) polymorphism and nucleotide sequences of representative G/G, G/A, and A/A genotypes are shown in Appendix A. Figure 1 shows representative increases in HbF following treatment with 30 nM MTH of ErPCs taken at different blood sampling times from different β-thalassemia patients.

While the ErPCs from the same patients exhibit the same trend of HbF induction (i.e., low HbF induction, as in the case of Fe57, or high HbF induction, as in the case of Fe28), the results obtained demonstrate greatly variable HbF inducibility for different β^0^/β^0^ and β^+^/β^+^ genotypes. From post hoc comparisons, no significant comparisons within genotypes (both within G/G and within G/A) are observed, while highly significant is the comparison between G/G and G/A genotypes (*p* < 0.0003), thus highlighting that the highest HbF induction was found in G/A rs368698783 genotypes.

### 2.2. Association between the rs368698783 Genotypes and Basal HbF Production by ErPCs

When basal levels of HbF were determined in ErPCs from β-thalassemia patients carrying G/G (n = 45), G/A (n = 12), and A/A (n = 3) genotypes, a significantly higher HbF production in cells from patients carrying G/A and A/A genotypes was demonstrated (Figure 2A–C).

In order to verify a possible link between the rs368698783 genotypes and HbF induction, in total, ErPCs from 38 β-thalassemia patients were in parallel induced with mithramycin, hydroxyurea, and rapamycin (for a final number of 57 independent induction experiments). In addition to these, ErPCs from another 16 patients were treated with mithramycin for an additional 39 independent induction experiments. The number of induction experiments was higher than the number of the 60 β-thalassemia patients analyzed because, in some cases, the induction experiments were repeated several times, obtaining similar results. The data obtained using the three HbF inducers are shown in Figure 2D–F.

Statistical values related to the HbF induced increase (average *±* SD) were the following: mithramycin (A/A: 41.29 ± 16.75; G/A: 16.88 *±* 12.21; G/G: 11.13 ± 12.83); hydroxyurea (A/A: 15.98 ± 13.84; G/A: 11.43 ± 5.12; G/G: 5.34 ± 5.67); and rapamycin (A/A: 30.33 ± 8.94; G/A: 21.22 ± 11.18; G/G: 6.59 ± 6.00).

As clearly appreciable (all ANOVA test results were highly significant), the highest induced increase in HbF production is obtained in ErPCs isolated from β-thalassemia carrying *Aγ-globin* gene rs368698783 G/A and A/A genotypes. In particular, in HU (Figure 2E) and RAPA (Figure 2F) induction, the differences when both G/A and A/A cohorts are compared to the G/G cohort, are highly significant, supporting the hypothesis that the “A” allele is dominant with respect to the “G” allele in driving the phenotype “induced HbF production”. This is confirmed by comparing, with the t-test on logarithmic data, the A/A plus G/A population with the G/G population, obtaining 0.0076, 0.0049, and <0.0001 *p* values for mithramycin, hydroxyurea, and rapamycin induction, respectively. Therefore, our data strongly suggest that the *Aγ-globin* gene rs368698783 (G>A) polymorphism should be considered among the markers for in vitro response to HbF inducers, at least for hydroxyurea and rapamycin.

### 2.3. Down-Regulation of LYAR mRNA in ErPCs Cultured with HbF Inducers

On the basis of these results, supporting a repressor role of LYAR in *γ-globin* gene transcription and in order to further analyze its role in the action of different HbF inducers, the expression of *LYAR* mRNA was quantified by RT-qPCR in ErPCs from 29 (for MTH induction) and 19 (for HU and RAPA induction) β-thalassemia patients (among which five β^0^-thalassemia, A/A or A/G and six β^+^-thalassemia G/G patients) treated with MTH, HU, and RAPA. The results depicted in Figure 3A give an indication for the down-regulation of *LYAR* mRNA together with the expected increase in induced *γ-globin* mRNA content, further supporting the role of LYAR in HbF induction. Interestingly, Figure 3B shows that *LYAR* mRNA down-regulation is more evident for all the inducers analyzed within the rs368698783 G/G β^+^-thalassemia ErPCs. This is particularly significant (*p* = 0.0256) for rapamycin treatment.

Accordingly, at least for some HbF inducers, down-regulation of LYAR might be required, unless in the case of attenuation of LYAR-associated activity on the *Aγ-globin* gene by the presence of rs368698783 G/A or A/A polymorphisms. The relationship and the role of LYAR in *γ-globin* gene expression will require correlation studies between LYAR protein accumulation and *γ-globin* gene transcription and the careful analysis of the effects of LYAR knockout or overexpression on HbF production, also in combination with HbF inducers acting through different mechanisms of action.

### 2.4. Increase in γ-Globin mRNA Is Associated with a Decrease in LYAR mRNA in ErPCs Isolated from Rapamycin-Treated β-Thalassemia Patients Participating in the NCT03877809 Clinical Trial

We took advantage of the ongoing NCT03877809 clinical trial in order to verify whether treatment in vivo of β-thalassemia patients with sirolimus is associated with the altered expression of the *LYAR* gene. During the clinical trial, β-thalassemia patients were treated with 1 mg/day of rapamycin for an increasing length of time (10, 90, 180, and 360 days). Results of this trial have been reported recently [23].

We found that the in vivo treatment with rapamycin induces an increase in *γ-globin* mRNA content in EPO-cultured ErPCs from the majority of β-thalassemia patients. In four patients, *LYAR* mRNA content was also analyzed and the data reported in Figure 4. As clearly evident and fully in agreement with the data shown in Figure 3B, an increase in *γ-globin* content is associated with a decrease in *LYAR* mRNA content (Figure 4). The results support the concept that *LYAR* gene expression is downregulated in vivo in ErPCs isolated from rapamycin-treated patients.

## 3. Discussion

Recently, Ju et al. [14] identified a putative novel regulator of *γ-globin* gene transcription, LYAR (human homologue of the mouse Ly-1 antibody reactive clone). The LYAR DNA-binding motif (5′-GGTTAT-3′) was identified in a DNA region corresponding to the 5′-untranslated region of the *Aγ-globin* gene and to the rs368698783 (G>A) polymorphism [15]. LYAR is a strong repressor of human *γ-globin* gene transcription, suggesting this is a novel transcription factor essential for *γ-globin* gene silencing [14]. Interestingly, we found that the rs368698783 (G>A) polymorphism is present in β-thalassemia patients [15,16]. The functional effect of the rs368698783 (G>A) polymorphism has been clarified by Bianchi et al. [15], who performed molecular docking simulations, an electrophoretic mobility shift assay (EMSA), and Biacore analyses demonstrating that the rs368698783 (G>A) polymorphism decreases the LYAR binding efficiency to the *Aγ-globin* gene and might affect cellular responses to HbF inducers [17]. Moreover, Chen et al. [24] have further explained the LYAR role in *Aγ-globin* gene transcription, showing that the rs368698783 (G>A) polymorphism triggers the attenuation of LYAR and two repressive epigenetic regulators, DNA methyltransferase 3 alpha (DNMT3A) and protein arginine methyltransferase 5 (PRMT5), from binding to the *Aγ-globin* gene. This mediates *γ-globin* increase by facilitating the demethylation of CpG sites in erythroid progenitor cells of β-thalassemia patients. The repressor function of LYAR with respect to *Aγ-globin* gene transcription explains the increased basal (Figure 2C) and induced (Figure 2D–F) levels of HbF in rs368698783 G/A and A/A β-thalassemia patients on the one hand, and the “dominant” characteristics of the “A” allele on the other.

The objective of this study was to stratify β-thalassemia patients with respect to the rs368698783 (G>A) polymorphism and to verify whether their erythroid precursors (ErPCs) differentially respond in vitro to selected HbF inducers. The location of this polymorphism and nucleotide sequences of the representative G/G, G/A, and A/A genotypes are shown in Appendix A. In order to perform this analysis on other β-thalassemia patients and different HbF inducers, ErPCs from an extended number of β-thalassemia patients carrying different rs368698783 genotypes were treated with mithramycin [19], hydroxyurea [20], and rapamycin [21,22,23]. The DNA-binding low molecular weight mithramycin (MTH) is one of the most powerful HbF inducers [19]. Hydroxyurea (HU) is currently employed by many research groups for the HbF induction in β-thalassemia and sickle-cell disease (SCD) patients [9]. The mTOR (mammalian Target of Rapamycin) inhibitor rapamycin (RAPA) is of great interest as a possible re-positioned drug for β-thalassemia and sickle-cell disease (SCD), as it is already used in patients undergoing kidney transplantation [10]. Rapamycin and rapamycin analogues induce HbF, as firmly established both in vitro on ErPCs and in vivo on experimental mice [25]. Moreover, Gaudre et al. [26] and Al-Khatti and Alkhunaizi [27] reported HbF induction in three kidney-transplanted SCD patients treated with everolimus [26] or rapamycin [27]. The possibility to propose mTOR inhibitors for in vivo pilot clinical studies is also sustained by the Orphan Drug Designation of rapamycin (as Sirolimus) by the European Medicinal Agency (EMA) (EU/3/15/1585) for the treatment of β-thalassemia. These actions were the basis of two clinical trials with rapamycin, NCT03877809 (A Personalized Medicine Approach for β-thalassemia Transfusion Dependent Patients: Testing sirolimus in a First Pilot Clinical Trial) and NCT04247750 (Treatment of β-thalassemia Patients with Rapamycin (Sirolimus): From Pre-clinical Research to a Clinical Trial) [22,23].

The results presented here could be considered important not only for basic biomedicine but also in applied translational research for precision medicine in personalized therapy of β-thalassemia. Despite the fact that for mithramycin and rapamycin, a correlation between the in vitro data using ErPCs and an in vivo response has not been explored so far, this has been reported in the case of hydroxyurea [28]. Accordingly, our data suggest that the rs368698783 polymorphism might be considered among the parameters useful to recruit patients with the highest probability of responding to in vivo treatment with HbF inducers. The high response to mithramycin of G/G β-thalassemia patients (see Figure 2D) might be explained by the fact that this DNA-binding drug is expected to exhibit a different mechanism of action. In particular, MTH might inhibit the interaction of several transcriptional repressors (such as Sp1) to regulatory sequences of the *LYAR* gene. EMSA experiments demonstrate that MTH inhibits in vitro molecular interactions between Sp1-complexes and Sp1 binding sites of the *LYAR* gene (Cosenza et al., unpublished results). In the case of rapamycin, this information is of great interest since this drug is presently employed in two clinical trials. When the relationship between rapamycin-induction of HbF and rs368698783 LYAR (G>A) polymorphism (high) was compared to the relationship with other HbF-associated polymorphisms, a high relationship was found, as expected, with XmnI [29], but a lower relationship was found to MYB rs9399137 [30], BCL11A rs14227407 [31], and BCL11A rs10189857 (Zuccato et al., unpublished results). A more extensive analysis, including other HbF-associate polymorphisms (such as KLF1), will be necessary to determine what is the polymorphism displaying the highest association with rapamycin-mediated HbF induction.

## 4. Materials and Methods

### 4.1. Recruitment of Patients, ErPCs Isolation, and ErPCs Induction

The patients were recruited, and the blood samples were obtained according to the Declaration of Helsinki following specific approvals of the study by the Ethical Committees of Ferrara Hospital. All the participants in the NCT03877809 (A Personalized Medicine Approach for β-thalassemia Transfusion Dependent Patients: Testing Sirolimus in a First Pilot Clinical Trial) signed informed consent on the basis of approvals of the Ethical Committee in charge of human studies at Arcispedale S.Anna, Ferrara (release of the approval: 14 November 2018). For ErPCs’ isolation and testing, the two-phase liquid culture procedure was employed as previously described. After 7 days of a phase I culture, the non-adherent cells were harvested, washed, and then cultured in a phase II medium in the absence or presence of HbF inducers. Control uninduced cells are ErPCs cultured in parallel with induced cultures but in the absence of the inducer. The solubilization reagent (MeOH/water 1:1, *v*/*v*) was unable to cause, at the concentration used, any changes in relative values and total amounts of hemoglobin production.

### 4.2. RNA Extraction from Erythroid Precursor Cells (ErPCs)

The total cellular RNA was extracted from the ErPCs by using TRI Reagent^®^ (Sigma-Aldrich, St. Louis, MO, USA), following the manufacturer’s instructions. The protocol used for the extraction of ErPCs RNA employed 800 μL of TRI Reagent^®^ for a dry pellet of 4–6 × 10^6^ cells, then chloroform and isopropanol volumes during the extraction were proportionally adjusted accordingly to the initial volume of TRI Reagent^®^ used; the isolated RNA was washed once with cold 75% ethanol, dried, and dissolved in 10–20 μL nuclease-free water before use.

### 4.3. RT-qPCR Analysis of Expression of Globin Genes and LYAR

For the gene expression analysis, 500 ng of total RNA was reverse transcribed by using the TaqMan^®^ Reverse Transcription Reagents and random hexamers (Applied Biosystems, Life Technologies, Carlsbad, CA, USA). A quantitative real-time PCR assay, to quantify the expression of the globin genes, was carried out using two different reaction mixtures, the first one containing a *γ-globin* probe and primers, the second one containing GAPDH, RPL13, and β-actin probes and primers. The primers and probes used are listed in Table 1.

Each reaction mixture contained 1× TaKaRa Ex Taq^®^ DNA Polymerase (Takara Bio Inc., Shiga, Japan), 300 nM of forward and reverse primers, and 200 nM of probes (Integrated DNA Technologies, Castenaso, Italy). The assays were carried out using the CFX96 Touch Real-Time PCR System (Bio-Rad, Hercules, CA, USA). After an initial denaturation at 95 °C for 1 min, the reactions were performed for 50 cycles (95 °C for 15 s and 60 °C for 60 s). Data were analyzed by employing the CFX manager software Ver. 3.1 (Bio-Rad, Hercules, CA, USA). To compare the gene expression of each template amplified, the ΔΔCt method was used [21,32]. The Lyar Assay was ID Hs00215132_m1, TaqMan assay 20x, Applied Biosystems, Thermo Fischer Scientific, Waltham, MA, USA.

### 4.4. HPLC Analysis of Hemoglobins

The ErPCs were centrifuged at 1200 rpm for 8 min and washed with PBS (Phosphate buffered saline). The pellets were then resuspended in a predefined volume of water for HPLC (Sigma-Aldrich, St. Louis, MO, USA). Then, three freeze/thaw cycles on dry ice were performed in order to lyse the cells and obtain the protein extracts. Hemoglobin analysis was performed by loading the protein extracts into a PolyCAT-A cation exchange column and then eluted in a sodium-chloride-BisTris-KCN aqueous mobile phase using HPLC Beckman Coulter instrument System Gold 126 Solvent Module-166 Detector, which allows us to obtain a quantification of the hemoglobin present in the sample. The reading is performed at a wavelength of 415nm, and a commercial solution of purified human HbAF (Analytical Control Systems Inc., Fishers, IN, USA) extracts has been used as the standard. The values thus obtained are processed using “32 Karat software”. The percent of HbF increase was identified by the following formula: 100 × (% HbF in induced cells − % HbF in control uninduced cells)/(100 − % HbF in control uninduced cells).

### 4.5. Statistical Analysis

All the data were normally distributed and presented as mean ± SD. Statistical differences between the groups were compared using one-way ANOVA (Analyses of Variance) between groups. Statistical differences were considered significant when *p* < 0.05 (*) and highly significant when *p* < 0.01 (**).

## 5. Conclusions

The results presented here are of interest not only for basic biomedicine but also in applied translational research for precision medicine in personalized therapy of β-thalassemia. In fact, our data suggest that the rs368698783 polymorphism might be considered among the parameters useful to recruit patients with the highest probability of responding to the in vivo treatment with HbF inducers. In addition, translational research studies should be considered to determine whether clinically therapeutic levels of HbF production can be achieved by targeting the LYAR *γ-globin* gene repressor.

## Figures and Tables

**Figure 1 ijms-24-00776-f001:**
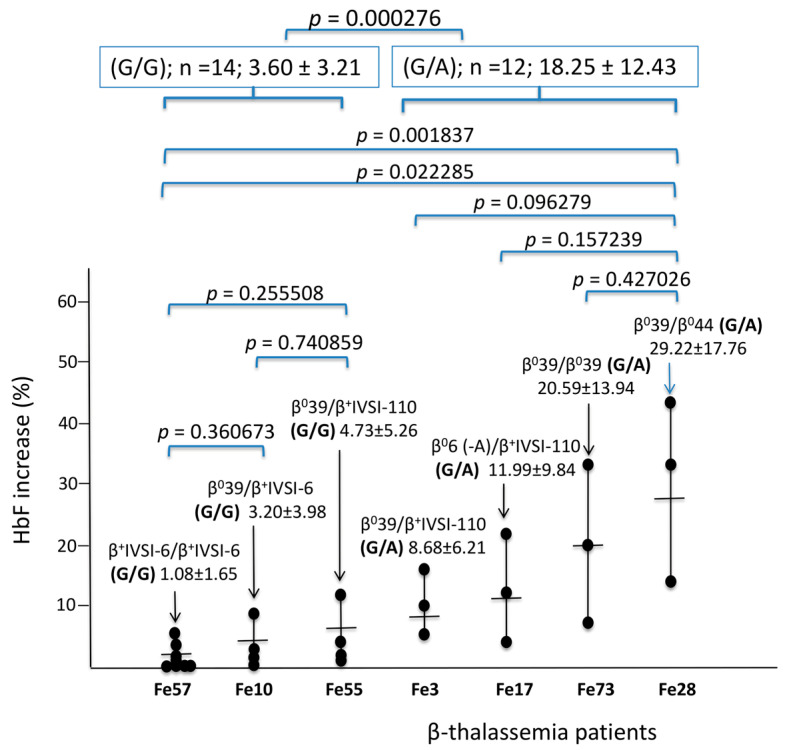
Heterogeneity in response to mithramycin (MTH) of ErPCs from different β-thalassemia patients recruited at Ferrara Hospital and accordingly labeled with “Fe”. The different dots represent independent MTH treatments on ErPCs isolated from the same patients at different times (at least three blood sampling for each of the indicated patients were performed at different times). ANOVA was used to test differences in the percent of HbF increase among genotypes; a post hoc LSD test was used, and the *p* values are indicated in each panel. “n” identifies the number of patients recruited. When more than one induction experiment was performed, each dot represents the average of the percent of HbF increase.

**Figure 2 ijms-24-00776-f002:**
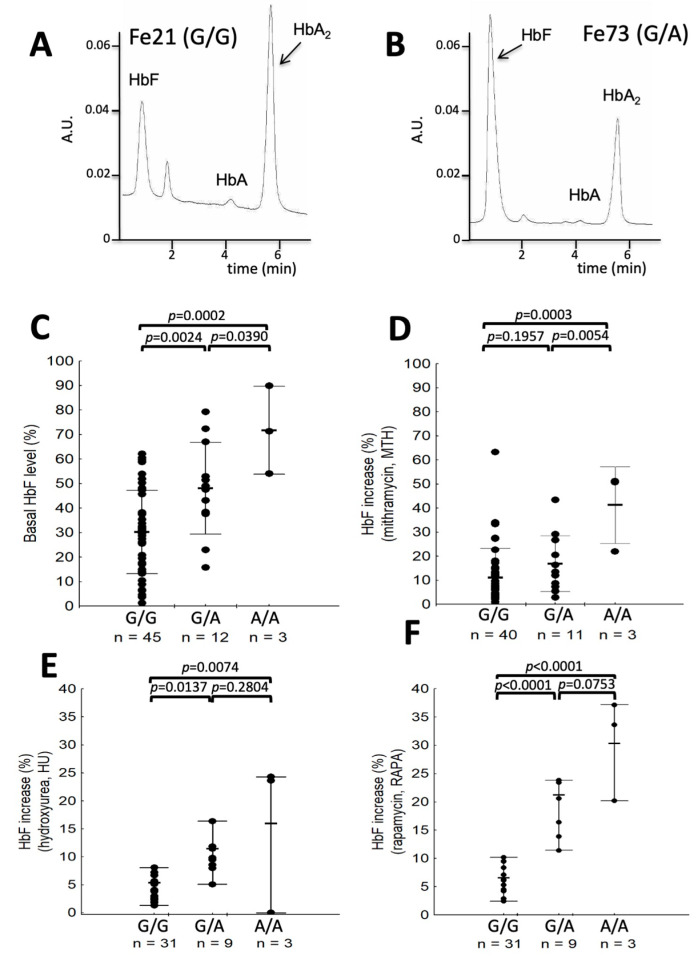
(**A**,**B**). Representative HPLC profile of ErPC lysates from β-thalassemia patients producing low (**A**) and high (**B**) HbF levels. (**C**) Basal levels of HbF in a total of 60 β-thalassemia patients. (**D**–**F**) HbF induction in ErPCs from G/G, G/A, and A/A β-thalassemia patients treated for 5 days in the presence of 30 nM mithramycin (**D**), 100 μM hydroxyurea (**E**), and 250 nM rapamycin (**F**). The percent of HbF increase was determined as described in the legend in Figure 1. ANOVA was used to test differences in the percent of HbF increase among genotypes; a post hoc LSD test was used, and the *p* values are indicated in each panel. “n” identifies the number of different patients recruited. When more than one induction experiment was performed, each dot represents the average of the percent HbF increase. The Shapiro–Wilk test was applied to analyze the data distribution of all quantitative traits. Logarithmic transformation was necessary either for baseline HbF levels (**C**) or for increased HbF levels after treatment with the three inducers (**D**–**F**). To test differences in the percent of HbF increase among genotypes, ANOVA was used on transformed data (**D**–**F**). A post hoc LSD test was used, and the *p* values are indicated in each panel in Figure 2. Averages and standard deviations of original data are shown for a straight understanding.

**Figure 3 ijms-24-00776-f003:**
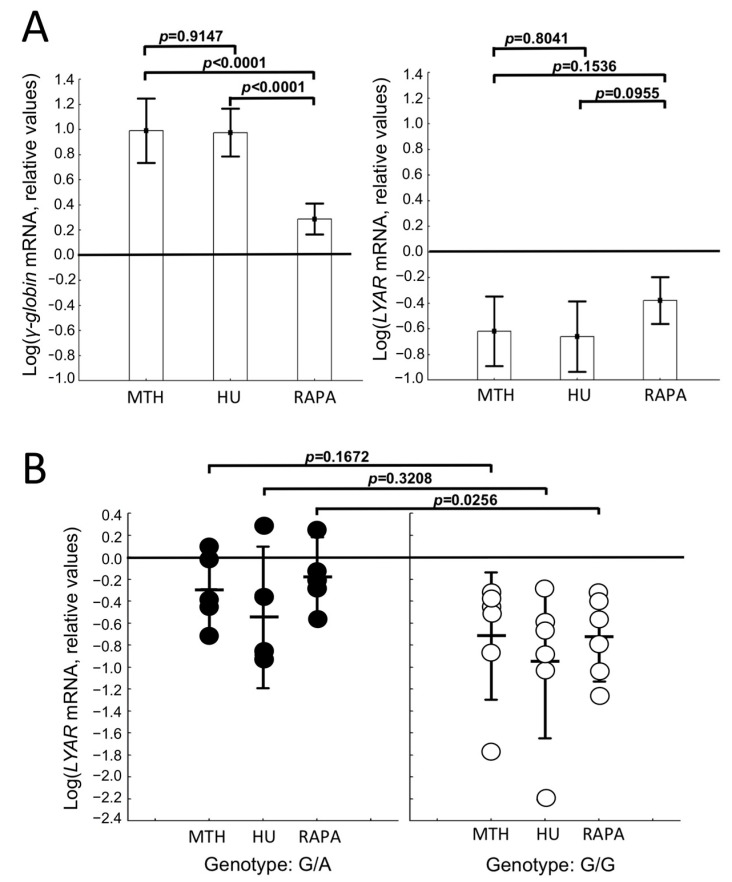
(**A**) Increase in *γ-globin* and decrease in *LYAR* mRNAs in 21 ErPC cultures (1 from A/A, 5 from G/A, and 15 from G/G rs368698783 polymorphism-carrying patients) treated for 5 days in the presence of 30 nM mithramycin, 100 μM hydroxyurea, and 250 nM rapamycin, as indicated. (**B**) Down-regulation of LYAR mRNA in the ErPCs reported in panel A but divided in A/A and A/G β^0^-thalassemia (all β^0^39 homozygous) and G/G β^+^-thalassemia (one β^+^IVSI-6/β^+^IVSI-6, three β^+^IVSI-110/β^+^IVSI-110, one β^+^IVSI-6/β^+^IVSI-110, and one β^+^-86/β^+^IVSI-110) ErPCs. ANOVA was used, on transformed data, to test differences, and *p* values from the post hoc LSD test are indicated in each panel.

**Figure 4 ijms-24-00776-f004:**
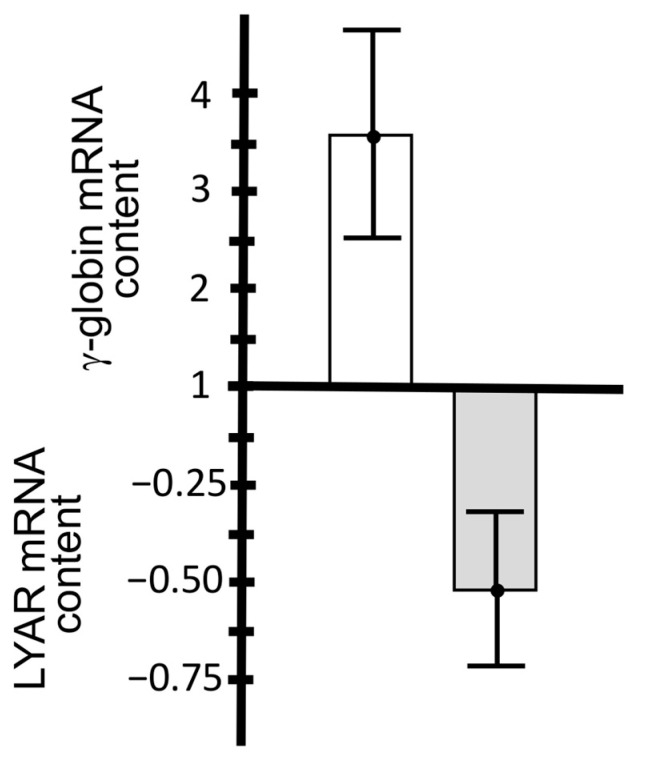
An increase in *γ-globin* mRNA is associated with a decrease in *LYAR* mRNA in ErPCs isolated from four β-thalassemia patients participating in the NCT03877809 clinical trial and treated for 90 days with 1 mg/day of rapamycin. Isolated ErPCs were treated with EPO in the absence of rapamycin.

**Table 1 ijms-24-00776-t001:** Sequences of the primers and probes employed.

Primer/Probes	Sequence
*γ-globin* forward (primer)	5′-TGACAAGCTGCATGTGGATC-3′
*γ-globin* reverse (primer)	5′-TTCTTTGCCGAAATGGATTGC-3′
*γ-globin* probe	5′-FAM-TCACCAGCACATTTCCCAGGAGC-BFQ-3′
RPL13A forward (primer)	5′-GGCAATTTCTACAGAAACAAGTTG-3′
RPL13A reverse (primer)	5′-GTTTTGTGGGGCAGCATCC-3′
RPL13A probe	5′-HEX-CGCACGGTCCGCCAGAAGAT-BFQ-3′
β-actin forward (primer)	5′-ACAGAGCCTCGCCTTTG-3′
β-actin reverse (primer)	5′-ACGATGGAGGGGAAGACG-3′
β-actin probe	5′-Cy5-CCTTGCACATGCCGGAGCC-BRQ-3′
GAPDH forward (primer)	5′-ACATCGCTCAGACACCATG-3′
GAPDH reverse (primer)	5′-TGTAGTTGAGGTCAATGAAGGG-3′
GAPDH probe	5′-FAM-AAGGTCGGAGTCAACGGATTTGGTC-BFQ-3′

## Data Availability

All the data related to the results presented will be made freely available upon request to the corresponding authors.

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
