# Peer review of "The rs368698783 (G>A) Polymorphism Affecting LYAR Binding to the Aγ-Globin Gene Is Associated with High Fetal Hemoglobin (HbF) in β-Thalassemia Erythroid Precursor Cells Treated with HbF Inducers"

_ijms, 2023, doi:10.3390/ijms24010776_

Round 1

Reviewer 1 Report

In this article, authors stratified ß-thalassemia patients with respect to the rs368698783 (G>A) polymorphism and investigated whether their erythroid precursors (ErPCs) differentially respond in vitro to selected HbF inducers, depending on the presence of the polymorphism investigated.

The manuscript is written clearly and concise. An improvement can be made considering the few comments below (mainly regarding article structure)

·         1. The paragraph from line 84 to 103 should be moved to discussion section

·         2. The paragraph from line 122 to 131 should be moved to M&M section under statistical analysis subsection (or to some extent in the Figure 2 caption)

·         3. The paragraph from line 144 to 149 should be moved to discussion section

·         4. The paragraph from line 159 to 170 should be moved to discussion section

Author Response

This the reply to the Reviewer #1.

General comments. In this article, authors stratified ß-thalassemia patients with respect to the rs368698783 (G>A) polymorphism and investigated whether their erythroid precursors (ErPCs) differentially respond in vitro to selected HbF inducers, depending on the presence of the polymorphism investigated. The manuscript is written clearly and concise. An improvement can be made considering the few comments below (mainly regarding article structure)

Answer. We thank the reviewer for her(his) positive comments and we hope that our changes will be considered satisfactory.

Point 1. The paragraph from line 84 to 103 should be moved to discussion section.

Answer. We moved this paragraph as suggested to the discussion section (pag 7, lines 218-237).

Point 2. The paragraph from line 122 to 131 should be moved to M&M section under statistical analysis subsection (or to some extent in the Figure 2 caption).

Answer. We moved this paragraph as suggested in the legends to Figures, when necessary (see for instance the red-marked portions of the legends to Figures 1 and 2.

Point 3. The paragraph from line 144 to 149 should be moved to discussion section.

Answer. We moved this paragraph as suggested to the discussion section (pag 7, lines 245-251).

Point 4. The paragraph from line 159 to 170 should be moved to discussion section.

Answer. We moved this paragraph as suggested to the discussion section (pag. 6-7, lines 200-213).

Reviewer 2 Report

This manuscript addresses the effect of rs368698783 (G>A) polymorphism on fetal hemoglobin (Hb F) in β-thalassemia erythroid precursor cells. Several comments as follows;

1. To verify the direct effect of rs368698783 (G>A) polymorphism on fetal hemoglobin expression, the genotypes of beta-thalassemia mutations, alpha-thalassemia, Hb F-associated SNPs (Xmn I, BCL11A, and HBS1L-MYB), and KLF1 mutation are suggested to provide in the manuscript.

2. Regarding manuscript writing, several sentences need to correct the position of the sentences i.e.

- Location of this polymorphism and nucleotide sequences of representative G/G, G/A and A/A genotypes are shown in Figure 1 (A and B) from Page 2, lines 66-67. This sentence should state in the result part and corrected in Figure S1.

-  There are several sentences with cited numbers of references in the result parts. This should rearrange to the correct position.

3. There are some issues with the quality of the figures and the descriptions of data in the figures also need to be significantly improved.

- Statistical analysis is required in Figure 1.

- Please check the correction of the label of Hb analysis by HPLC, especially Hb A2 and Hb A.

Author Response

This the reply to the Reviewer #2.

General comments. This manuscript addresses the effect of rs368698783 (G>A) polymorphism on fetal hemoglobin (Hb F) in β-thalassemia erythroid precursor cells. Several comments as follows.

Point 1. To verify the direct effect of rs368698783 (G>A) polymorphism on fetal hemoglobin expression, the genotypes of beta-thalassemia mutations, alpha-thalassemia, Hb F-associated SNPs (Xmn I, BCL11A, and HBS1L-MYB), and KLF1 mutation are suggested to provide in the manuscript.

Answer. The referee is right. Unfortunately, the number of HbF related polymorphisms is very high and deserves future experimental work. We are performing these studies and we confirm that the LYAR polymorphisms, together with the XmnI, is that displaying the highest association with HbF induction. We are very much interested in this issue, as we are running two clinical trials using rapamycin/sirolimus) as HbF inducer. We followed the point raised by the referee, by including the sentence ( “When the relationship between rapamycin-induction of HbF and rs368698783 LYAR (G>A) polymorphism (high) was compared to the relationship with other HbF-associated polymorphisms high relationship was found, as expected, with XmnI [29], but lower relationship was found to MYB rs9399137 [30], BCL11A rs14227407 [31] and BCL11A rs10189857 [31] (Zuccato et al., unpublished results). A more extensive analysis including other HbF-associate polymorphisms (such as KLF1) will be necessary to determine what is the polymorphism displaying the highest association with rapamycin-mediated HbF induction”), page 7, lines 253-257.

Point 2. Regarding manuscript writing, several sentences need to correct the position of the sentences i.e.: Location of this polymorphism and nucleotide sequences of representative G/G, G/A and A/A genotypes are shown in Figure 1 (A and B) from Page 2, lines 66-67. This sentence should state in the result part and corrected in Figure S1.

Answer. We checked sentences for correct placing and moved the sentence mentioned by the referee to the results section (page 2, lines 70-71). We slightly changed the supplementary materials accordingly.

Point 3. There are several sentences with cited numbers of references in the result parts. This should rearrange to the correct position.

Answer. We checked all sentences and relative references to identify possible misplaced sentences and/or citations. All the (minor) changes have been red-marked.

Point 4. There are some issues with the quality of the figures and the descriptions of data in the figures also need to be significantly improved.

Answer. We tried to improve the description of the data presented in the Figures (changes red marked throughout). Concerning the quality of the Figures, we have tried to enlarge them as much as possible and are available to solve specific issues with the MDPI production teams, in the case the manuscript will be accepted.

Point 5. Statistical analysis is required in Figure 1.

Answer. Thank for this observation. We included statistical analysis in Figure 1 and briefly commented it in the text.

Point 6. Please check the correction of the label of Hb analysis by HPLC, especially Hb A2 and Hb A.

Answer. Thank for this observation. We performed the required check. In particular all the data shown in Figure 2 appear to be correct.

Round 2

Reviewer 2 Report

All comments have been addressed in the revised manuscript. However, please remove the cited numbers in the objective and result parts (page no. 2). Cited number of references can state in the introduction or discussion parts.

Author Response

Sammi Wang
Section Managing Editor
E-Mail: [email protected]

Dear Dr. Wang,

Thank you for your letter and for the comments by the reviewer, confirming that all comments were addressed by us in the revised manuscript (R1).

According to the final suggestions of the reviewer, we have removed the cited numbers of references in the Results section. Some key references were cited, as suggested in the final part of Introduction (lines 64-67). All the changes have been red-marked in this revised version (R2).

As required, we have also verified that all the references were necessary and we checked again their numbering.

We hope that with these further changes, the paper will be considered acceptable and we thank you and the referees for your work, very useful for helping us in improving the scientific quality and presentation of our study.

Sincerely,

Prof. Alessia Finotti

Department of Life Sciences and Biotechnology

Ferrara University
